



# Compilation of Southern Ocean sea-ice records covering the last glacial-interglacial cycle (12-130 ka)

Matthew Chadwick[1], Xavier Crosta[2], Oliver Esper[3], Lena Thöle[4], Karen E. Kohfeld[5,6]

[1]British Antarctic Survey, Cambridge, UK
[2]UMR 5805 EPOC, Université de Bordeaux, CNRS, EPHE, Pessac, France
[3]Alfred Wegner Institute, Helmholtz Centre for Polar and Marine Research, Bremerhaven, Germany
[4]Department of Earth Sciences, Utrecht University, Utrecht, the Netherlands
[5]School of Resource and Environmental Management, Simon Fraser University, Vancouver, Canada
[6]School of Environmental Science, Simon Fraser University, Vancouver, Canada

*Correspondence to*: Matthew Chadwick (machad27@bas.ac.uk)

**Abstract.** Antarctic sea ice forms a critical part of the Southern Ocean and global climate system. The behaviour of Antarctic sea ice throughout the last glacial-interglacial (G-IG) cycle (12,000-130,000 years) allows us to investigate the interactions between sea ice and climate under a large range of mean climate states. Understanding both temporal and spatial variations in Antarctic sea ice across a G-IG cycle is crucial to better understanding the G-IG regulation of atmospheric $CO_2$, ocean

circulation, nutrient cycling and productivity. This study presents published qualitative and quantitative estimates of G-IG sea ice from twenty four marine sediment cores, and an Antarctic ice core. Sea ice is reconstructed from the sediment core records using diatom assemblages and from the ice core record using sea-salt sodium flux. Whilst all regions of the Southern Ocean display the same overall pattern in G-IG sea-ice variations, the magnitudes and timings vary between regions. Sea-ice cover is most sensitive to changing climate in the output regions for the Weddell Sea and Ross Sea Gyres, as indicated by the greatest

magnitude changes in sea ice in these areas. In contrast the Scotia Sea sea-ice cover is much more resilient to moderate climatic warming, likely due to the meltwater stratification from high iceberg flux through 'iceberg alley' helping to sustain high sea-ice cover outside of full glacial intervals. The differing sensitivities of sea ice to climatic shifts between different regions of the Southern Ocean has important implications for the spatial pattern of nutrient supply and primary productivity, which subsequently impact carbon uptake and atmospheric $CO_2$ concentrations changes across a G-IG cycle.

## 1 Introduction

Antarctic sea ice is a crucial component of the Southern Hemisphere and global climate system, through a strong albedo feedback (Hall, 2004), its influence on ocean-atmosphere gas exchange (Rysgaard et al., 2011), its roles in deep and bottom water formation (Rintoul, 2018) and consequently oceanographic circulation (Maksym, 2019). Sea ice also helps to stabilise Antarctic ice shelves and marine terminating ice streams by buffering against wave and ocean swell induced calving (Massom

et al., 2018). Over the last four decades, Antarctic sea-ice extent has shown a slight increasing trend, although within the last five years are the two years with the lowest annual sea-ice extent in the observational record (Parkinson, 2019). The causes



and drivers of this observed sea-ice extent trend are not well understood, with numerous potential mechanisms invoked (Bintanja et al., 2013; Ferreira et al., 2015; Lecomte et al., 2017; Turner et al., 2016). Model simulations have been unable to replicate sea-ice changes during the observational period without an unrealistically reduced warming trend (Rosenblum and

Eisenman, 2017). In order to better understand sea-ice changes and diagnose the response of the ocean system to these changes at the multi-decadal timescale we must investigate the regional patterns, magnitudes and timings of Antarctic sea-ice changes throughout a full glacial-interglacial (G-IG) cycle. Investigating the last G-IG cycle will allow us to document the interactions between sea ice and climate during both very different mean climate states, such as the warmer-than-present last Interglacial, and transient climate states, such as glacial terminations or inceptions.

Sea ice has been linked with changes in carbon sequestration, ocean circulation and productivity across G-IG cycles (Kohfeld et al., *this volume*). Over the last glacial-interglacial cycle (12-130 ka) atmospheric $CO_2$ has been observed to decrease in a stepwise fashion, with at least three major intervals of $CO_2$ drawdown at 115,000-100,000 years (115-100 ka), 72-65 ka and 40-18 ka (Jouzel et al., 2007). These intervals correspond to Marine Isotope Stage (MIS) or sub-stage boundaries (MIS 5e-5d, MIS 5a-4 and MIS 3-2, respectively). These changes in atmospheric $CO_2$ have been tied to several mechanisms operating in

the Southern Ocean, including: (a) reductions in air-sea gas exchange, (b) enhanced stratification of surface waters, (c) reduced upwelling and vertical diffusion, (d) deep-ocean circulation changes, and (e) changes in marine biological productivity and nutrient cycling. None of these mechanisms are mutually exclusive, and all can be linked in some way to changes to Antarctic sea-ice cover. Specifically, sea ice regulates deep ocean outgassing through both surface stratification and by acting as a physical barrier (Rysgaard et al., 2011). Brine rejection during sea-ice formation densifies Antarctic bottom waters, thus

increasing water column stratification and reducing vertical mixing (Bouttes et al., 2010; Ferrari et al., 2014; Galbraith and de Lavergne, 2019). Both surface and deep stratification of the water column enhance $CO_2$ sequestration in Southern ocean abyssal waters. Finally, sea-ice cover in the Southern Ocean influences marine primary productivity, and thus $CO_2$ uptake, both by acting as a barrier to light and through the release of nutrients, especially iron, in meltwaters within the marginal ice zone (Arrigo et al., 2008). Changes in Southern Ocean sea-ice extent across a G-IG cycle have therefore the potential to play

a key role in atmospheric $CO_2$ variability (Kohfeld and Chase, 2017; Peacock et al., 2006). As a result, understanding the timing and magnitude of sea-ice changes in the Southern Ocean is a crucial question for understanding the G-IG regulation of ocean circulation, productivity, nutrient cycling, and carbon uptake.

Diatom assemblages preserved in marine sediments currently provide the most robust proxy for investigating past changes in Antarctic sea ice over long time intervals (Crosta et al., *this volume;* Thomas et al., 2019). Sea ice can be reconstructed from

past diatom assemblages both qualitatively, using the relative abundance of sea-ice related diatoms, and quantitatively, using statistical transfer functions. In this study we have compiled both qualitative and quantitative published sea-ice proxy records from twenty four Southern Ocean marine sediment cores, alongside one Antarctic ice core sea-ice proxy record (Figure 1), for the last 150 ka, with particular focus on the 12-130 ka interval, to answer the following questions:



- Did different regions of the Southern Ocean display the same patterns of sea-ice advance and retreat across the last G-IG cycle?
- Was the timing and magnitude of sea-ice changes during the last G-IG cycle consistent between different Southern Ocean regions?
- Was sea ice more resilient/sensitive to climatic shifts in the different regions of the Southern Ocean?

## 2 Materials and methods

### 2.1 Core sites

We have compiled twenty eight sea-ice proxy records from twenty four marine sediment cores and the EPICA Dome C (EDC) ice core (Table 1). Data are presented for the 0-150 ka age range (Figure 1) but the focus of this study is the 12-130 ka range, as this interval covers a single G-IG cycle from the midpoint of Termination II to the midpoint of Termination I. The data from 0-12 ka and 130-150 ka is included to give additional insights into Southern Ocean sea-ice behaviour during other interglacial and glacial periods. Of the twenty seven sea-ice proxy records from marine sediment cores, fourteen of them cover the full 12-130 ka interval and the other thirteen only cover part of the interval (Table 1). The cores presented in this study are located between 43 and 62 °S (Figure 1) and the majority are located in the oceanographic region between the modern winter sea-ice extent (WSIE) and the modern Antarctic Polar Front (APF). Almost all diatom-based sea-ice records covering the last G-IG cycle are restricted to the north of the mean WSIE. Retrieving adequate cores south of the mean WSIE is difficult for two reasons. Firstly, heavy sea-ice cover limits biogenic silica production. Secondly, most of this region overlies abyssal plains with depths greater than 4000 m. As a result of both low production and a long settling time, dissolution of the more lightly silicified diatom species (generally sea-ice related species) increases, which biases the preserved diatom assemblage to reflect warmer and lower sea-ice conditions.

We selected records that stretch back beyond 30 ka, to ensure that they cover more than just MIS 2, and also excluded short term or 'snapshot' reconstructions (e.g. the Last Glacial Maximum synthesis by Gersonde et al. (2005) or the MIS 5e synthesis by Chadwick et al. (2020)). Any records with a mean sample resolution of >3 ka were also excluded from this synthesis. To avoid additional chronological uncertainties from converting between multiple different age models (e.g. Capron et al. (2014)), the chronologies for all the records in this study are kept as originally published. Therefore, the records presented in this study have a chronological uncertainty of ~2-4 ka (Bazin et al., 2013b; Lisiecki and Raymo, 2005; Parrenin et al., 2007; Veres et al., 2013), which is not significant compared to the timescale we focus on.

**Table 1: Details for the twenty four sediment cores and one Antarctic ice core which have sea-ice proxy records presented in this study, including location, temporal coverage and sample resolution of the data, what form the reconstructed sea-ice data comes in and references for both the data and chronology for each record. Cores are ordered by latitude from north to south. FCC: combined relative abundance of the diatom species *Fragilariopsis curta* and *F. cylindrus*, MS: magnetic susceptibility, Na_{ss}: sea-salt sodium, SID: sea-ice duration, SST: sea-surface temperature, WSIC: winter sea-ice concentration.**



| | Latitude (°S) | Longitude (°E) | Temporal coverage (ka) | Sample resolution (ka): mean (min-max) | Sea-ice reconstruction type | Data references | Chronology |
|---|---|---|---|---|---|---|---|
| SK200/22a | 43.70 | 45.07 | 2-95 | 0.8 (0.1-2.1) | SID | Nair et al. (2019) | Radiocarbon ages combined with correlation of the planktonic and benthic $\delta^{18}$O to Antarctic ice cores (Byrd & EDML) and the LR04 $\delta^{18}$O stack (Manoj and Thamban, 2015). |
| PS2499-5 | 46.51 | -15.33 | 0-130 | 1.4 (0.1-4.1) | FCC | Gersonde and Zielinski (2000) | Radiocarbon ages and $^{230}$Th excess dating combined with biofluctuation stratigraphies for *Eucampia antarctica* and *Cycladophora davisiana* and planktonic $\delta^{18}$O compared to SPECMAP (Gersonde and Zielinski, 2000). |
| SK200/27 | 49.00 | 45.22 | 1-73 | 0.3 (0.1-1.3) | SID | Nair et al. (2019) | Radiocarbon ages combined with correlation of the planktonic and benthic $\delta^{18}$O to Antarctic ice cores (Byrd & EDML) and the LR04 $\delta^{18}$O stack (Manoj and Thamban, 2015). |
| PS1778-5 | 49.01 | -12.70 | 12-134 | 2.3 (0.3-14) | FCC | Gersonde and Zielinski (2000) | Radiocarbon ages and $^{230}$Th excess dating combined with biofluctuation stratigraphies for *Eucampia antarctica* and *Cycladophora davisiana* and planktonic $\delta^{18}$O compared to SPECMAP (Gersonde and Zielinski, 2000). |
| ODP 1093 | 49.98 | 5.87 | 11-150 | 2.7 (1.1-6.3) | FCC | Schneider Mor et al. (2012) | Correlation of transfer function summer SSTs, *N. pachyderm* $\delta^{18}$O and MS to $\delta$D and dust records from the EDC ice core (Schneider Mor et al., 2012). |



| | | | | | | |
|---|---|---|---|---|---|---|
| PS1768-8 | 52.59 | 4.48 | 3-146 | 1.6 (0.4-2.6) | WSIC | Esper and Gersonde (2014) | Radiocarbon ages and regional correlation of diatom species composition, estimated summer SSTs and MS between proximal cores (Xiao et al., 2016a) combined with $^{230}$Th excess dating (Frank et al., 1996). |
| PS2102-2 | 53.07 | -4.99 | 0-35 <br> 72-133 | 1 (0.3-4) | WSIC <br> FCC | Xiao et al. (2016a) <br> Bianchi and Gersonde (2002) | Radiocarbon ages combined with regional correlation of diatom species composition, estimated summer SSTs and MS between proximal cores (Xiao et al., 2016a). <br><br> *N. pachyderma* $\delta^{18}$O changes compared to SPECMAP combined with diatom biofluctuation zones (Bianchi and Gersonde, 2002). |
| ODP 1094 | 53.18 | 5.13 | 0-150 | 2.4 (0.3-4.8) | FCC | Schneider Mor et al. (2012) | Correlation of transfer function summer SSTs, *N. pachyderma* $\delta^{18}$O and MS to $\delta$D and dust records from the EDC ice core (Schneider Mor et al., 2012). |
| TN057-13-PC4 | 53.20 | 5.10 | 0-46 | 0.9 (0.5-1.9) | SID | Stuut et al. (2004) | Radiocarbon ages (Shemesh et al., 2002). |
| PS2606-6 | 53.23 | 40.80 | 1-36 | 0.4 (0.1-0.8) | WSIC | Xiao et al. (2016a) | Radiocarbon ages combined with regional correlation of diatom species composition, estimated summer SSTs and MS between proximal cores (Xiao et al., 2016a). |
| PS1652-2 | 53.66 | 5.10 | 1-33 | 1.8 (0.7-3.7) | WSIC | Xiao et al. (2016a) | Radiocarbon ages combined with regional correlation of diatom species composition, estimated summer SSTs and MS between proximal cores (Xiao et al., 2016a). |
| TPC063 | 53.92 | -48.04 | 0-46 | 0.4 (0.1-4) | FCC | Collins et al. (2013) | Correlation of relative paleointensity to the SAPIS RPI stack combined with *E. antarctica* biofluctuation stratigraphy (Collins et al., 2012). |





| PS2276-4 | 54.64 | -23.57 | 0-130 | 1.8 (0.5-3.5) | FCC | Gersonde and Zielinski (2000) | Radiocarbon ages and $^{230}$Th excess dating combined with biofluctuation stratigraphies for *Eucampia antarctica* and *Cycladophora davisiana* and planktonic $\delta^{18}$O compared to SPECMAP (Gersonde and Zielinski, 2000). |
|---|---|---|---|---|---|---|---|
| SK200/33 | 55.02 | 45.15 | 6-150 | 1.5 (0.1-5) | WSIC & SID | Ghadi et al. (2020) | Radiocarbon ages combined with the correlation of SST and SID to the temperature record from the EDC ice core (Ghadi et al., 2020). |
| PS67/197-1 | 55.14 | -44.11 | 4-30 30-86 | 0.4 (0.2-0.7) | WSIC FCC | Xiao et al. (2016a) Xiao (2011) | Radiocarbon ages and regional correlation of Scotia Sea ash layers combined with MS correlation to the EDML ice core dust record, diatom abundance fluctuation patterns and recorded geomagnetic excursions (Xiao et al., 2016b). |
| TPC078 | 55.55 | -45.02 | 1-31 | 1.4 (0.7-2.5) | FCC | Collins et al. (2013) | Radiocarbon ages combined with MS correlation to EDC ice core dust (Collins et al., 2012). |
| SO136-111 | 56.67 | 160.23 | 3-150 | 1.1 (0.6-2.2) | SID | Crosta et al. (2004) | Radiocarbon ages combined with the correlation of *N. pachyderma* $\delta^{18}$O to the SPECMAP reference stack (Crosta et al., 2004). |
| PS67/219-1 | 57.22 | -42.47 | 5-30 30-150 | 1 (0.2-10.1) | WSIC FCC | Xiao et al. (2016a) Xiao (2011) | Radiocarbon ages and regional correlation of Scotia Sea ash layers combined with MS correlation to the EDML ice core dust record, diatom abundance fluctuation patterns, recorded geomagnetic excursions and the Last Common Occurrence of *Rouxia leventerae* (Xiao et al., 2016b). |



| | | | | | | | |
|---|---|---|---|---|---|---|---|
| PS75/072-4 | 57.56 | -151.22 | 1-150 | 1.1 (0.4-13.7) | FCC | Studer et al. (2015) | Radiocarbon ages and the correlation of $\delta^{18}$O to the LR04 stack combined with regional correlation of elemental composition and MS between proximal cores (Benz et al., 2016). |
| TAN1302-96 | 59.09 | 157.05 | 2-140 | 1.8 (0.4-6.5) | WSIC & SID | Jones et al. (2022) | Radiocarbon ages combined with the correlation of *N. pachyderma* $\delta^{18}$O to the LR04 stack (Jones et al., 2022). |
| ELT27-23 | 59.62 | 155.24 | 2-52 | 1 (0.1-4.9) | WSIC | Ferry et al. (2015a) | Radiocarbon ages combined with the correlation of *N. pachyderma* $\delta^{18}$O to the LR04 stack (Ferry et al., 2015a). |
| TPC034 | 59.79 | -39.60 | 0-62 | 1.8 (0.7-3.3) | FCC | Allen et al. (2011) | *E. antarctica* and *C. Davisiana* biofluctuation stratigraphies combined with correlation of the MS to EDC ice core dust (Allen et al., 2011). |
| PS58/271-1 | 61.24 | -116.05 | 6-148 | 0.6 (0.2-1.6) | WSIC | Esper and Gersonde (2014) | Correlation of physical parameters, elemental composition, diatom assemblage composition and derived WSIC and SSTs to the EDC ice core record combined with diatom biostratigraphic data (Esper and Gersonde, 2014). |
| TPC286 | 61.79 | -40.14 | 0-95 | 2 (0.2-21.9) | FCC | Collins et al. (2013) | Correlation of relative paleointensity to the SAPIS RPI stack combined with *E. antarctica* biofluctuation stratigraphy (Collins et al., 2012). |
| EDC | 75.10 | 123.35 | 1-150 | 2 (2-2) | Na$_{ss}$ Flux | Wolff et al. (2006) | Glaciological ice flow model constrained by dated volcanic horizons and matching of isotopic variations to other ice core records (Parrenin et al., 2007). |





### 2.2 Sea-ice reconstructions

In this study, we present both quantitative and qualitative reconstructions of sea ice (Figure 1). Quantitative reconstructions are either the winter sea-ice concentration (WSIC) or the sea-ice duration (SID), with two cores (SK200/33 and TAN1302-96) that have both a WSIC and SID record. All but one (core ELT27-23) of the quantitative reconstructions use a Modern Analog Technique (MAT) diatom transfer function, with cores PS1768-8, PS2102-2, PS2606-6, PS1652-2, PS67/197-1, PS67/219-1 and PS58/271-1 run using the MAT transfer function detailed in Esper and Gersonde (2014) and cores SK200/22a, SK200/27,

TN057-13-PC4, SK200/33, SO136-111 and TAN1302-96 run using the MAT transfer function detailed in Crosta et al. (1998) and their evolutions. Core ELT27-23 uses the Generalised Additive Model transfer function detailed in Ferry et al. (2015b). A comparison of reconstructed sea-ice values using both the MAT and Generalised Additive Model transfer functions shows that both techniques yield robust and broadly similar results (Ferry et al., 2015b) and supports our inclusion in this study of records produced with either technique.

For cores where no quantitative sea-ice reconstructions were available we have used a qualitative reconstruction in the form of the combined relative abundance of the diatom species *Fragilariopsis curta* and *F. cylindrus* (FCC). The FCC proxy is a qualitative indicator of winter sea-ice (WSI) presence, with abundances >3 % associated with locations south of the mean WSI edge (Gersonde and Zielinski, 2000). The sea-salt sodium ($Na_{ss}$) flux in the EDC ice core is also a qualitative indicator of WSIE, with a higher $Na_{ss}$ flux associated with a greater WSIE (Wolff et al., 2010).

Alongside the raw sea-ice data, we have also normalised all twenty eight records (Figure 2) to allow for a comparison of the timing of major changes in sea ice between records. We used the following normalisation formula: $(X_{sample} - X_{mean})/(X_{max} - X_{min})$, where the mean, max and min are calculated using all available data within the 0-150 ka interval for each record. Alongside the individual normalised sea-ice records we also present a stack of normalised records for five different regions of the Southern Ocean (Figures 2 and 4). In order to stack the normalised records for each region, the individual normalised

records were first resampled at 2 ka resolution. This age resolution was chosen as a compromise between maintaining the variation in each record and minimising the amount of interpolation required.

Whilst the normalised records allow us to investigate changes in the timing of major sea-ice changes between records they do not allow a comparison of the magnitude of sea-ice changes. For this comparison we have standardised the quantitative sea-ice records (Figure 3) using the following formulas: $(X_{WSIC} - 20)/100$ and $(X_{SID} - 1)/10$. The 20 % and 1 month/yr values in the

two formulas represent the mean WSIE, north of which sea ice occurs only episodically. This means x-axis values in the Figure 3 graphs can indicate the relative position of the mean WSI edge, with positive x-axis values indicating periods when the mean WSI edge is located north (equatorward) of the core site and negative values indicating when the mean WSI edge is to the south (poleward) of the core site. The 100 % and 10 months/yr in the two formulas are scaling factors that show how the WSIC or SID compares to a theoretical maximum. In the modern day, SIDs <8 months/yr have a good linear correlation to WSICs





(Figure S1). Given all the SID values in our records are <5 months/yr we have used a theoretical maximum of 10 months/yr SID to keep this scaling against WSICs, rather than the actual maximum of 12 months/yr SID. The FCC and Na$_{ss}$ flux proxies are only qualitative indicators of sea ice and cannot be scaled in the same way as the WSIC and SID records. Records with these qualitative proxies are thus excluded from the map of standardised records (Figure 3).

To allow for comparison of sea-ice patterns and trends between different regions of the Southern Ocean the records in Figures

1 and 2 are grouped into: the Scotia Sea (60 – 30 ºW), the Central Atlantic (30 – 0 ºW), the East Atlantic (0 – 30 ºE), the West Indian (30 – 60 ºE) and the Pacific (150 ºE – 105 ºW) sectors. These sectors were chosen to allow us to investigate longitudinal variations in sea ice, whilst ensuring there are at least four core records in each sector. In Figure 3 the Central and East Atlantic sectors are combined into a single Atlantic sector (20 ºW – 30 ºE) due to the smaller number of standardised records.

### 2.3 Principal component analysis

A Principal Component Analysis (PCA) was applied to the resampled and normalised data using PAST v3.15 software (Hammer et al., 2001) in order to identify the main trends present in the twenty eight marine-based sea-ice records (from twenty four sites) and the EDC Na$_{ss}$ flux record. We applied the PCA to the normalised data to avoid over-representation of a given proxy or record (e.g. FCC varies between 0 and 20 % in our records whereas Na$_{ss}$ flux varies between 200 and 1000 µg/m$^2$/yr). The PCA was performed over the entire 0-150 ka period. Many of the records do not cover the whole period (Figure

1) and missing values were coped with by using the mean value imputation option.

**Figure 1: FCC, WSIC and SID records from Southern Ocean marine sediment cores and the Na$_{ss}$ flux record from the EDC ice core shown alongside a map of Antarctica with core locations, modern February (blue line) and September (black line) sea-ice extents (data from Fetterer et al. (2017)) and the modern APF position (grey line; Trathan et al. (2000)). Downcore records are arranged**
**together in Southern Ocean regions, with distance from the modern WSI edge given in degrees latitude (+ indicates locations north of the mean WSI edge, - indicates locations south of the mean WSI edge).**

**Figure 2: Normalised FCC, WSIC and SID records from Southern Ocean marine sediment cores and the Na$_{ss}$ flux record from the EDC ice core shown alongside a map of Antarctica with core locations, modern February (blue line) and September (black line) sea-ice extents (data from Fetterer et al. (2017)) and the modern APF position (grey line; Trathan et al. (2000)). Downcore records are**
**arranged together in Southern Ocean regions, with distance from the modern WSI edge given in degrees latitude (+ indicates locations north of the mean WSI edge, - indicates locations south of the mean WSI edge). Each region also has a stacked record with the mean (black line) and interquartile range (grey shading) shown.**

**Figure 3: Standardised WSIC and SID records from Southern Ocean marine sediment cores shown alongside a map of Antarctica with core locations, modern February (blue line) and September (black line) sea-ice extents (data from Fetterer et al. (2017)) and**
**the modern APF position (grey line; Trathan et al. (2000)). Downcore records are arranged together in Southern Ocean regions, with distance from the modern WSI edge given in degrees latitude (+ indicates locations north of the mean WSI edge, - indicates locations south of the mean WSI edge). Unlike in Figures 1 and 2, records from the East and Central Atlantic sectors have been combined into a single Atlantic sector region.**

**Figure 4: Comparison of the stacks of normalised sea-ice records from the five Southern Ocean sectors (black curves mark the**
**means and grey shading marks the interquartile ranges) alongside the stack of normalised Southern Ocean sea-surface temperature (SST) records (the blue curve marks the mean and the blue shading marks the interquartile range), the normalised LR04 benthic δ$^{18}$O stack (purple curve; Lisiecki and Raymo (2005)) and the normalised atmospheric CO$_2$ record from the EDC ice core (red**





curve; Bazin et al. (2013a)). **The selection of the SST data is detailed in Section S1. Vertical dashed lines mark MIS boundaries, with each MIS numbered in the centre of the figure.**

## 3 Results and discussion

### 3.1 Patterns and trends in sea ice

The PCA of all the Southern Ocean sea-ice records in this study reveals that the dominant trend, principal component (PC) 1, in sea ice during the last 130 ka is the G-IG cyclicity, with high sea-ice cover during MIS 2 and low sea ice during MIS 5 (Figure S2). PC1 explains 40 % of the total variance, while PC2 accounts for only ~10 % (Figure S3) and if the PCA is applied solely to the long records, which cover over half of the 0-150 ka interval, PC1 accounts for >50 % of the variance and PC2 is reduced to 8 %. All PCs except PC1 fall at or below the broken stick curve (Frontier (1976); Figure S3) and therefore, only PC1 appears to bear enough signal to be interpreted. The G-IG trend in PC1 is also evident in the normalised stacks for the different sectors of the Southern Ocean, with the exception of the Scotia Sea sector where there is high sea ice throughout the 0-100 ka period (Figures 2 and 4). The PCA does not produce any grouping based on the different methods of sea-ice reconstruction used (i.e. FCC, WSIC, SID) (Figure S4), which suggests that all three estimates of sea ice display similar patterns and supports our decision to include all three in this study. There is also no apparent PCA grouping by Southern Ocean sector, although the two most southerly Scotia Sea sector cores (TPC034 and TPC286) do have a very different trend to all the other records (Figure S4).

All Southern Ocean sectors in this study have the same pattern in sea-ice retreat, with a prominent drop during the glacial termination at 140-130 ka, but the pattern in sea-ice advance varies between the sectors (Figures 1, 2 and 4). The West Indian and Pacific sectors have low sea ice from the glacial termination through until ~80-70 ka, after which the normalised sea-ice values display a gradual increase to maxima at 25-20 ka (Figures 2 and 4). The East Atlantic sector also has low sea ice until ~70 ka but this is followed by a rapid increase to high normalised sea-ice values during MIS 4 and then a retreat at the beginning of MIS 3, although not as pronounced as during MIS 5 (Figure 4). Normalised sea-ice values in the East Atlantic then increase gradually from ~55 ka until a maximum concurrent with the other Southern Ocean sectors (Figure 4). Both the Central Atlantic and Scotia Sea sectors have sea ice expansion to glacial or near-glacial levels in the normalised values by 100 ka. Normalised sea-ice values in the East Atlantic reach a maximum during the 30-20 ka period in phase with other sectors, except in the Scotia Sea where normalised sea-ice values do not present much variation over the last 100 ka. As such, the pattern in Central Atlantic sector sea ice indicates a transition between the normalised sea-ice trends in the East Atlantic and Scotia Sea sectors (Figures 2 and 4). The EDC $Na_{ss}$ flux record has an almost bimodal pattern in values, with both rapid decreases and increases between the interglacial and glacial 'states' (Figures 1 and 2). Between 70 and 20 ka there is almost no variation in the EDC $Na_{ss}$ flux whereas between 130 and 110 ka there is almost a quadrupling of the EDC $Na_{ss}$ flux (Figure 1). This is in contrast to the trends seen in the sediment core sea-ice records (Figures 1, 2 & 4) and was suggested by Röthlisberger et al. (2010) to indicate that the $Na_{ss}$ flux in Antarctic ice cores is more sensitive to sea-ice changes during interglacial intervals than during





200  glacial intervals. However, it should be noted that the EPICA Dronning Maud Land (EDML) ice core (75.00 ºS, 0.07 ºE) Na$_{ss}$
flux record does show more variation during the 70-20 ka interval (Fischer et al., 2007), which suggests that sea-ice cover may
be more dynamic in the Weddell Sea than around the Wilkes Land margin, in agreement with the results in Ghadi et al. (2020).

MIS 5 substages can be seen in both the EDC record and several of the Pacific and Atlantic sector core records (PS1778-5,
PS1768-8, PS75/072-4, SO136-111 and TAN1302-96), whereas the Indian sector records maintain low sea ice from
205  Termination II through until MIS 4 (Figure 1). The prominence of these substages in the normalised records for cores
PS75/072-4 and PS1768-8 (Figure 2) is likely due to these cores being located near the output regions of the Ross Sea and
Weddell Sea Gyres, respectively. Both the raw and normalised FCC records from core PS67/219-1 also appear to show MIS
5 substages but with an offset of ~5 ka to younger ages compared to the other records (Figures 1 and 2).

The standardised sea-ice records indicate that the majority of cores with WSIC or SID reconstructions were located within the
mean WSIE during MIS 2, with the exception of the most northerly West Indian sector cores (Figure 3). Standardised sea-ice
records also show that, despite sea-ice expansion in most regions as early as MIS 4 (Figures 1, 2 and 4), the mean WSI edge
was located south of the majority of core sites in this study until MIS 2 (Figure 3). The standardised sea-ice record for core
PS58/271-1 shows high frequency, but low amplitude, variability around the y-axis for the majority of the interval from 110
to 30 ka (Figure 3), indicating that the mean WSI edge was located at or near this core site throughout this period.

**3.2 Magnitude of sea-ice changes**

The standardised records show that the largest amplitude changes in sea ice are found in the core records located at the output
of the Weddell Sea Gyre (Figure 3), indicating that the sea ice in this region is the most sensitive to climate variations. The
higher sensitivity of sea ice at the Weddell Gyre output than other regions of the Southern Ocean supports the greater sea-ice
advance seen in this region in studies of the Last Glacial Maximum (Gersonde et al., 2005) as well as the substantial retreat
estimated by model runs of the Last Interglacial (Holloway et al., 2017). The output regions of Southern Ocean gyres are areas
of high sea-ice drift in the present day, with changes in Southern Hemisphere winds strongly influencing this drift (Holland
and Kwok, 2012; Kwok et al., 2017). As discussed in Section 3.1, cores located at the output of the Southern Ocean gyres also
show larger normalised sea-ice variations during the colder MIS 5 substages (5b and 5d) than other cores (Figure 2), further
indicating the sensitivity of sea ice in these regions to lower amplitude climatic shifts than full G-IG transitions. Many of the
cores in this study, especially those in the West Indian and western Pacific sectors, show relatively little sea-ice signal outside
of the higher values during MIS 2-4 (Figure 1), suggesting that these cores are located too far north to pick up small magnitude
changes in sea ice.

In the East and Central Atlantic sectors, the normalised stacks indicate a substantial reduction in sea ice during MIS 3 (Figure
4). For the Central Atlantic sector stack this trend is largely driven by the normalised sea-ice values in core PS1778-5 (Figure
2), which is the most easterly Central Atlantic sector core site for which an MIS 3 record is presented (Figure 2). This behaviour



in the two Atlantic sectors contrasts to the normalised stacks for the other Southern Ocean sectors, where high sea ice is maintained throughout the entire MIS 2-4 interval (Figure 4). In the Pacific sector the only records to show low sea ice during MIS 3 are those from core TAN1302-96, in clear contrast with the nearby SO136-111 and ELT27-23 core records (Figure 1). However, the lower sea ice during MIS 3 in core TAN1302-96 compared to SO136-111 is attributed to a lower sample

resolution in the former (Jones et al., 2022). The reduction in sea ice during MIS 3 in the Central and East Atlantic, but not in the other Southern Ocean regions, raises intriguing questions regarding the spatial differences in Antarctic sea-ice sensitivity. Additional records in these other Southern Ocean sectors, at more southerly latitudes, are necessary to assess the robustness of the different sea-ice patterns during MIS 4-3 and the underlying drivers.

The Scotia Sea records indicate an enlarged sea-ice extent, relative to the modern, almost throughout the last 150 ka, with the

only exception being during MIS 5e (Figure 1). The high raw and standardised WSIC values in cores PS67/197-1 and PS67/219-1 suggest that the WSI edge was still located ~5 ° further north than its present-day location in the Late Holocene at ~4 ka (Figures 1 and 3). The location of these cores in the modern day 'iceberg alley' (Weber et al., 2014) could account for the pervasive sea-ice presence and low sensitivity to climatic variations. A maintained flux of icebergs through this region would release cold and fresh meltwaters, stratifying the surface ocean and promoting sustained sea-ice presence, even during

warmer periods. The notable reduction during MIS 5e in the Scotia Sea could therefore either indicate a change in the pathway or flux of Weddell Sea icebergs, or sufficiently warm sea-surface temperatures, maybe as a result of a more southerly position of the Antarctic Circumpolar Current in the Drake Passage and Scotia Sea region (Wu et al., 2021), to outweigh the influence of the iceberg meltwaters. The large amplitude changes in sea ice at the Weddell Sea Gyre output, as discussed above, are also likely related to 'iceberg alley', with an easterly expansion of the modern iceberg field (Weber et al., 2014) during colder

glacial periods providing greater iceberg meltwater stratification in the East Atlantic sector, which promoted and maintained an increased WSIE.

### 3.3 Timings of sea-ice changes

The different chronologies applied to the records in this study (Table 1) prevent us from being able to identify any small (~2-4 ka) differences in the timing of sea-ice changes between the different Southern Ocean regions. However, during both

Termination I and II, we observe up to 5 ka offsets in the retreat time of sea ice between the different regions (Figure 4). For both the Pacific and West Indian sectors, the normalised stack shows that sea-ice retreat during Termination II started at ~140 ka and was finished by ~130 ka, largely coincident with the increases in both the Southern Ocean SSTs and the atmospheric $CO_2$ record from the EDC ice core (Figure 4). In contrast, the normalised stacks for both the Central and East Atlantic sectors display sea-ice retreat between ~135 ka and ~125 ka, consistent with the timing of Termination II in the LR04 benthic $\delta^{18}O$

stack (Figure 4). In the Scotia Sea sector, the only record covering Termination II is from core PS67/219-1 and indicates a longer period of sea-ice retreat lasting from ~140 ka until ~125 ka (Figures 1 and 4). The earlier retreat in the West Indian sector compared to the two Atlantic sectors is likely due to the West Indian sector cores being located further north of the



modern winter sea-ice extent than the Atlantic sector cores, with an average of 9.2 degrees of latitude compared to 3.8 degrees, respectively (Figure 1). However, this does not explain the earlier sea-ice retreat in the Pacific sector, where the cores are located an average of only 3.7 degrees of latitude north of the modern winter sea-ice extent (Figure 1).

During Termination I, the Pacific sector normalised stack again displays an earlier sea-ice retreat than the other regions, with retreat starting at ~21 ka compared to ~18 ka in the East Atlantic and West Indian sectors and ~16 ka in the Central Atlantic sector (Figure 4). Whilst the timing of Termination I in the East Atlantic and West Indian sectors occurs within chronological uncertainty of Termination I in either the Central Atlantic or the Pacific sector, these latter regions differ by >4 ka in their timings for Termination I. The onset of Termination I sea-ice retreat in the East Atlantic, Central Atlantic and West Indian sectors is concurrent with the onset of Southern Hemisphere surface air temperature increases, which dominantly begin at ~17 ka (Osman et al., 2021), and increases in Southern Ocean SSTs (Figure 4). The earlier sea-ice retreat during Termination I in the East Atlantic sector normalised stack, compared to the Central Atlantic sector stack, is likely due to the proximity of the eastern core sites to the Weddell Sea Gyre output. Gersonde et al. (2005) reconstructed a 'tongue' of summer sea ice in this region during the Last Glacial Maximum which suggests sea ice in this area is more dynamic and may have been more susceptible to early melting and retreat during Termination I than the sea ice located further west. The discrepancy between the earlier retreat of sea ice in the East Atlantic sector, relative to the Central Atlantic sector, during Termination I but not Termination II could indicate that either the sea-ice 'tongue' in the East Atlantic sector was more resilient to melting and retreat during MIS 6 than during MIS 2, or that this 'tongue' did not exist during MIS 6. The high sensitivity of East Atlantic sector sea ice during Termination I is also indicated by the rapid retreat to interglacial levels, with a deglaciation duration of only ~2-4 ka compared to the ~5-8 ka deglaciation duration in the other Southern Ocean sectors (Figure 4). The normalised stack for the Scotia Sea does not show any clear signal for Termination I (Figure 4), with only cores TPC063 and TPC078 displaying the sharp decrease in sea ice between 20 and 12 ka that characterises Termination I in the other records (Figure 1).

## 4 Summary and outstanding questions

The present compilation of sea-ice records suggests that all regions of the Southern Ocean display the same general pattern in sea ice during the last G-IG cycle, with high sea ice during MIS 2 and 4 and low sea ice during MIS 5e. However, the different areas of the Southern Ocean do have varying magnitudes, timings and short-term trends in sea ice during the 12-130 ka interval, especially with regards to sea-ice advance between MIS 5e and 4. Sea-ice cover seems to be most sensitive to changing climate at the outputs of the Weddell Sea and Ross Sea Gyres, with the greatest magnitude sea-ice changes occurring in cores located in these regions. Records located near the Weddell Sea Gyre output have especially high sensitivity, with a more rapid sea-ice advance and retreat than other areas of the Southern Ocean and a more pronounced MIS 3 sea-ice reduction than the other regions. In contrast, the sea-ice cover in the Scotia Sea region is seemingly much more resilient to changing climate, with the only notable reduction during the peak of the MIS 5e interglacial. This resilience to moderate warming could be related to the





large iceberg flux through the Scotia Sea, with iceberg-meltwater stratification and cooling helping to sustain sea ice outside
of full glacial intervals.

By compiling long term sea-ice records of the last G-IG cycle from throughout the Southern Ocean this study has identified
how changes in sea ice vary between different regions of the Southern Ocean. The more dynamic state of sea ice in the East
Atlantic over long time scales is likely associated with the expansion and contraction of 'iceberg alley'. Weddell Sea icebergs
have been identified as an important source of Fe to the Atlantic sector of the Southern Ocean (Shaw et al., 2011) and large
amplitude changes in the spatial extent of iceberg outflow over G-IG cycles therefore helps explain the high magnitude
variations in G-IG Fe flux in this region of the Southern Ocean (Martínez-Garcia et al., 2009). Large amplitude changes in
both the East Atlantic sector WSIE and the spatial extent of 'iceberg alley' over a G-IG cycle have important implications for
the supply of nutrients, especially Fe, and subsequently primary productivity in this region. This, in turn, helps regulate oceanic
$CO_2$ sequestration across a G-IG cycle, in an area of the Southern Ocean with high modern primary productivity (Vernet et al.,
305 2019).

Our compilation also allows us to identify where further research would most help to develop the results and findings of this
study. It is clear from the discussion in section 3.3 that establishing a harmonised chronology across all core records could
help reduce the ~2-4 ka chronological uncertainties presented in this study and allow for further analysis of any short duration
leads and lags between the sea-ice changes in the different Southern Ocean sectors. Alongside an improved chronology, N-S
transects of sea-ice records, especially in the regions with the highest amplitude of G-IG sea-ice variability (e.g. East Atlantic
sector), would allow the rates of sea-ice advance and retreat to be estimated. A refined chronology would also allow sea-ice
dynamics to be compared between hemispheres and provide insights into how sea ice influences global ocean circulation
dynamics across a G-IG cycle.

Model-proxy comparisons of sea-ice extent over the last 130 ka display mixed results, with good agreement between model
and proxy estimates of WSIE during the Last Glacial Maximum (Green et al., 2021, *in review*) but poor model-data consistency
in estimates of summer sea-ice extent during the Last Glacial Maximum (Green et al., 2021, *in review*) and WSIE during the
Last Interglacial (Chadwick et al., *submitted;* Otto-Bliesner et al., 2021). Antarctic sea-ice changes during the observational
period have also not been replicated in model simulations, without an unrealistically reduced warming trend (Rosenblum and
Eisenman, 2017). Better parameterisation of sea ice (and assimilation of proxy records) is required to improve our
understanding of the drivers and feedbacks active over both short and long timescales.

Finally, whilst this study presents a good spatial coverage in some areas of the Southern Ocean (e.g. the Atlantic sector), there
are parts of the Southern Ocean for which G-IG sea-ice changes are minimally constrained (Figures 1-3). The central and
eastern Pacific sector (90 - 180 ºW) currently has only two long G-IG sea-ice records and the eastern Indian sector (60 - 150
ºE) has none. Similarly, the majority of the G-IG sea-ice records presented here are located north of the modern mean WSIE

(Figures 1-3), largely due to the difficulties associated with reconstructing robust sea-ice estimates from diatom records located beneath heavy sea-ice cover (as discussed in section 2.1). The development of new sea-ice proxies from beneath heavy sea ice (e.g. highly branched isoprenoids (Belt, 2018; Lamping et al., 2021; Vorrath et al., 2019) or DNA from sea-ice associated organisms (De Schepper et al., 2019)) could allow the reconstruction of sea ice from more southerly core sites, without which any small amplitude sea-ice variations during warmer interglacial periods (e.g. MIS 5a, 5c and 5e) cannot be fully investigated.

**Data availability**

The sea-ice proxy data for all twenty eight marine sediment core records are available from PANGAEA (doi pending, Chadwick et al. (*in review*)).

**Author contributions**

MC – Conceptualisation, Data curation, Formal analysis, Investigation, Methodology, Visualisation, Writing – original draft;
XC – Conceptualisation, Formal analysis, Methodology, Resources, Supervision, Visualisation, Writing – review & editing; OE – Conceptualisation, Resources, Writing – review & editing; LT – Data curation, Investigation, Resources; KEK – Conceptualisation, Funding acquisition, Investigation, Project administration, Supervision, Writing – review & editing.

**Competing interests**

The authors declare they have no conflict of interest.

**Acknowledgements**

This work was conducted as part of Phase 1 of the Cycles of Sea-Ice Dynamics in the Earth system (C-SIDE) PAGES scientific working group and was supported by a PAGES Data Stewardship Scholarship (DSS_105) to MC and a National Sciences and Engineering Research Council of Canada (NSERC) Discovery Grant (#RGPIN342251) to KEK. We thank both Claire S. Allen and Jacob Jones for directly providing us with their sea-ice proxy records from cores TPC034 and TAN1302-96, respectively.

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





Figure 1





Figure 2



Figure 3

Figure 4

