# Peer review of "Compilation of Southern Ocean sea-ice records covering the last glacial-interglacial cycle (12-130 ka)"

_Climate of the Past, 2022_

## Author Response (AR1)

**Reviewer 1 Response**

We thank the reviewer for their supportive and constructive comments on our manuscript. Please see below, in blue, our detailed responses to the comments.

This is a well-written paper that compiles sea ice records from 24 sediment cores around Antarctica covering the past 130 ka. The contribution is novel and important as it covers a much longer time frame than previous compilations. The authors use a creative method for comparing disparate types of records by normalizing the winter sea ice, annual sea-ice duration, and sea-salt sodium records to identify periods of higher or lower sea ice at each site. This method, coupled with record stacking, allows the to make comparisons about sea ice expansion and retreat on glacial-interglacial and finer timescales. The authors are careful not to overinterpret changes in timing, keeping in mind the error behind the cores' age models, but they are able to discern important differences between the 5 sectors of the Southern Ocean, noting both that the output regions of the Weddell and Ross Sea gyres are more sensitive to changes in sea ice and also that the Pacific and Indian sectors lead the Atlantic in sea ice retreat during Termination II, which the Pacific sector leads all others during Termination I.

However, there are a few areas where additional clarification is needed both in the text and in the figures. I detail the major issues below followed by my line-by-line comments.

When I read the abstract, I was not entirely sure what the authors meant by "output regions." It is eventually defined in section 3.2, but the term is used several times before this section. This could be clarified by maybe referring to these regions as "regions of high sea ice outflow" and/or including the ocean currents on at least one of the figures.

We have amended the text to clarify what we mean by output regions and have included the main Weddell Sea and Ross Sea gyre circulation patterns in Figure 1.

*"Sea-ice cover is most sensitive to changing climate in the regions of high sea-ice outflow from the Weddell Sea and Ross Sea gyres,…"*

It is unclear why the PCA figures are only included as supplemental information. I'm guessing it is because it doesn't add much information beyond the fact that glacial/interglacial cyclicity is the dominant forcing? It might be useful to plot PC1 along with the stacked sea ice records in Fig. 4.

The reviewer is correct in the reasoning behind us only including the PCA figures in supplemental. However we have added PC1 to Figure 4.

In section 3, there is a great deal of attention paid to the differences between MIS 5 through MIS 2. However, I had a really hard time seeing the MIS 5 substages referenced in lines 203-208. Perhaps some delineation of the plots in Figs. 1-3 would aid in this (i.e., colored bars indicating glacial/interglacial stages and substages). The changes during the MIS 5 substages are significantly more subtle than the difference between MIS 5 and MIS 4. The authors also refer to "the prominence" of the substages in Fig. 2 for cores PS75/072-4 and PS1768-8. However, for PS1768-8 there is only one small increase in sea ice around 100 ka. This does not depict the prominence of MIS 5b and 5d. PS1778- is much more prominent in Fig. 2. Is there a typo?

*We agree that the MIS 5 substages in PS1768-8 are not overly prominent in Figure 2 and have rephrased the text to refer to Figure 1, where the MIS 5 substages are more obvious for this core. We have also removed the word prominent from the sentence.*

It would be helpful if the authors briefly reiterated that positive values in Fig. 3 (note, Fig. 3 axes are unlabelled) indicate times when the cores were south of the winter sea ice extent in the past.

*We have restated this point to make it clearer to the reader:*

*"For the standardised records in Figure 3, positive values indicate intervals when the core site is located south of the mean WSI edge."*

Lines 211-212 state, "Standardised sea-ice records also show that, despite sea-ice expansion in most regions as early as MIS 4 (Figures 1, 2 and 4), the mean WSI edge was located south of the majority of core sites in this study until MIS 2 (Figure 3)." However, only the W. Indian sector agrees with this statement. When I look at Fig. 3, I see that many cores do not extend back to MIS 4. There is no information in the Scotia Sea sector about MIS 4, In the Atlantic sector, the only core that extends to MIS 4 shows that the site was south of the WSI edge in MIS 6 and MIS 4. In the Pacific Sector, two sites show similar magnitude expansions of sea ice in MIS 4 and MIS 2 (SO136-111 and the SID in TAN1302-96), whereas the WSI estimate in TAN1302-96 shows that the site was south of the WSI edge in MIS 6 and MIS 2. While, as the paper rightly points out PS58/271-1 shows highly fluctuating sea ice and was probably close to the WSI margin. I ask the authors to re-evaluate and revise their quoted statement above.

*We have amended the text to clarify that there are relatively few records that extend back to MIS 4. We have also specified the different patterns for the individual sectors, as mentioned by the reviewer:*

*"Most regions experience sea-ice expansion as early as MIS 4 (Figures 1, 2 and 4) but identifying the position of the mean WSI edge prior to MIS 4 is complicated by the scarcity of standardised records for this period (Figure 3). In the West Indian sector the mean WSI edge was located south of all the core sites in this study until MIS 2 (Figure 3). In the Pacific sector the SO136-111 and TAN1302-96 core sites were located north of the mean WSI edge during the 130-70 ka interval (MIS 5), with the standardised record for TAN1302-96 indicating it was north of the mean WSI edge during the 130-25 ka interval (MIS 5 through MIS 3, inclusive). In contrast, the standardised sea-ice record for core PS58/271-1…"*

Line 225: This sentence is confusing because the authors say that cores from the western Pacific sector show little sea ice signal outside of MIS 2-4, but the western Pacific cores (PS75/072-4; SO136-111; TAN1302-96) are the ones that earlier were described as having variability in the MIS 5 substages. It's unclear whether this is a typo and the authors meant the Scotia Sea or if there's something about these western Pacific cores that is not obvious here.

*The earlier reference has been amended to remove cores SO136-111 and TAN1302-96 from the discussion of MIS 5 substages.*

In line 241, the authors estimate how much farther north the WSI edge was in the Scotia Sea during MIS 2. It took me a while to parse that this was probably estimated because

there is a lot of sea ice in the cores during MIS 2 and they are currently up to 3.6 degrees north of WSI. It would be helpful if a line or two describing this logic could be included in the text. While on the topic of the Scotia Sea, I also wonder if the sediments below iceberg alley artificially show increased sea ice because sea ice diatoms are transported by the flux of ice bergs?

We have added additional text to clarify the logic behind our estimation:

*"The high raw and standardised WSIC values in cores PS67/197-1 and PS67/219-1 indicate that these cores were located south of the mean WSI edge in the Late Holocene at ~4 ka (Figures 1 and 3). This suggests that the Late Holocene WSI edge in the Scotia Sea was located ~5 ° further north than its present day location."*

It is possible that sea-ice diatoms are being transported with icebergs and artificially inflating our reconstructions of sea ice. However, this is not seen in the core top diatom abundances in Chadwick et al. (2022) and we would expect that substantial lateral transport of sea-ice diatoms would result in signs of break-up and dissolution, which are not apparent in the Holocene sediments in the Scotia Sea cores (Xiao et al. 2016).

**Short Line by Line Comments:**

Lines 80-81: I was confused by this, and initially thought that the sites north of the WSIE also should be in abyssal depths. I looked up a bathymetric map of Antarctica and was surprised to learn that there is a network of ridges north of Antarctica, roughly in the same location as the WSIE boundary. I suggest mentioning the water depth of cores in lines 76-79 somewhere (and potentially in Table 1?) to underscore that the cores mostly come from shallower depths.  It's not necessary for this paper, but if you're interested in a modern analysis of sea ice extent and water depth, Nghiem et al., 2016; doi:10.1016/j.rse.2016.04.005 is quite interesting. I found this paper while deciphering Lines 80-81.

We have added a comment on the average water depth for the core records presented.

Lines 82-83: The authors write, "dissolution of the more lightly silicified diatom species (generally sea-ice related species) increases, which biases the preserved diatom assemblage to reflect warmer and lower sea-ice conditions." This is a commonly written idea, but in my experience, finding actual data to back this up is challenging. I urge the authors to find a reference to support this idea of biased diatom assemblages (i.e. sea ice diatoms selectively dissolved) and/or increased dissolution of diatoms in general.

Leventer (1998) and Warnock et al. (2015) both discuss the dissolution of diatoms causing biased reconstructions and are now referenced in the manuscript in support of the sentence identified by the reviewer.

Line 147: Journal requirements differ, but as a reader, I would appreciate to be reminded of what the acronyms FCC, WSIC, SID, APF, and WSI stand for in this figure caption.

We have spelled out the acronyms in the figure captions.

Line 217: It's unclear which cores are in the output region of the Weddell Sea Gyre. Is it the Atlantic Sector cores? Please include a notation.

We have added a note with which cores are in the output region of the Weddell Sea Gyre.

Lines 296-305: I suggest you spell out iron throughout this paragraph.

We have refered to it as iron throughout the paragraph.

Lines 314-320: This paragraph relies fully on non-peer-reviewed papers. If the Green et al., 2021 and Chadwick, et al submitted papers are In Press by the time this manuscript is submitted, this paragraph is fine (in fact it's great). But I just wanted to highlight it in case they're not.

The Green et al. paper is now published.

**Figures**:

**Figures 1-3:**

Please label major places referred to in the test including (but not limited to) Ross Sea, Weddell Sea, Scotia Sea…it would also be helpful to include very basic current patterns since you mention output regions repeatedly. It is very difficult to discern the difference between the grey and black lines (September sea ice vs Antarctic Polar Front). Perhaps the grey line could be lighter?

We have added labels of the Ross, Weddell and Scotia Seas and the general gyre patterns to Figure 1. We have also made the difference between the grey and black lines more distinct.

Each of the records is numbered in Figs. 1-3, however the numbers change from figure to figure. I realize that this is because the number of records decreases, however, it makes it confusing to compare between figure. I suggest the authors number the records in Fig. 1 and leave the numbers consistent even though it means that Fig. 3 will not have record numbers 1, 3, 5, 6…

We have kept the numbers consistent with Figure 1 throughout.

In the paper you refer to studies as reconstructing winter sea ice concentration and abbreviate it WSIC, but in the figures you call it WSI (%). Please make these consistent.

We have changed the figure legends to WSIC (%).

**Figures 2 and 3** do not have the x axis defined. Is it the same for every core? It should be. Is it +/- 1? Even though it's normalized, it should still be labelled. Actually, in the legend, it says that the axes are still WSI (%), SID (months/year), etc. But, if this data is normalized, shouldn't it be unitless?

The x axis is consistent between all the records and hasn't been labelled because the values themselves are not relevant to the interpretation. The values should be unitless and we have made this change.

**Figure 4:** Please label Terminations I and II.

We have added labels for Terminations I and II.

**Supplementary Material**

Figures S5 and S6 are not referred to in the text. Please either include a discussion of them or remove them from the supplemental material.

We have added a mention of Figures S5 and S6 in the main text when discussing the SST compilations.

Chadwick M., Allen C.S., Sime L.C., Crosta X. & Hillenbrand C.-D. 2022. How does the Southern Ocean palaeoenvironment during Marine Isotope Stage 5e compare to the modern? *Marine Micropaleontology*, **170**: 102066.

Leventer A. 1998. The fate of Antarctic "sea ice diatoms" and their use as paleoenvironmental indicators. In: *Antarctic Sea-ice, Biological Processes, Interactions and Variability*, Lizotte M.P. & Arrigo K.R. Eds., Antarctic Research Series, American Geophysical Union, Washington D.C. **73:** 121-137.

Warnock J.P., Scherer R.P. & Konfirst M.A. 2015. A record of Pleistocene diatom preservation from the Amundsen Sea, West Antarctica with possible implications on silica leakage. *Marine Micropaleontology*, **117**: 40-46.

Xiao W., Esper O. & Gersonde R. 2016. Last Glacial - Holocene climate variability in the Atlantic sector of the Southern Ocean. *Quaternary Science Reviews*, **135**: 115-137.

We thank the reviewer for their supportive and constructive comments on our manuscript. Please see below, in blue, our detailed responses to their comments.

Chadwick et al. present a valuable paper on the sea ice evolution around Antarctica, based on diatom records from a variety of previous publications. The paper is generally well written, though there are a few points that need clarifying, in my opinion. I recommend publications after these points have been addressed.

 **Major comments:**

- Using diatoms as sea ice proxies: given that diatoms reproduce in the spring/summer, I am wondering how one the main reconstructed variable from the proxy records is winter sea-ice concentration? I think it would be good to explain this in more detail in the methodology to understand the reasoning behind this notion.

Whilst the main flux of sea-ice diatoms to the seafloor does occur during spring/summer, Gersonde & Zielinski (2000) showed that sea-ice diatoms are being exported to the seafloor throughout the year, not solely during spring/summer. As the winter sea-ice melts during spring, nutrients and meltwater are released, creating a nutrient-rich stratified surface layer, which promotes the growth of diatom blooms, including many species that are seeded directly from the sea ice. Spring blooms of diatoms in this marginal ice zone produce the winter sea-ice signal in an assemblage averaged for an entire year. The greater the winter sea-ice concentration, the shorter the open ocean season between spring melt back and autumn refreezing. In this situation, open ocean diatoms are much less competitive than sea-ice associated diatoms because they lack anti-freeze proteins (Janech et al. 2006, Bayer-Giraldi et al. 2011). Increased relative and absolute abundances in sediment traps (Gersonde & Zielinski 2000) and surface sediments (Zielinski et al. 1998) have been observed southwards, along with increased sea-ice concentrations, as a direct response of the greater relative dominance of the marginal ice zone bloom compared to purely open ocean species in the annual assemblage.

The use of diatom assemblages to reconstruct past winter sea-ice concentrations is robust and well documented in numerous published manuscripts and therefore a detailed explanation of the justification behind this proxy is not within the scope of this manuscript.

- There is a general confusion with how many records are in this compilation. The abstract says 24 sediment cores (+ one ice core). The Materials and Methods section says 28 sea-ice proxy records from 24 sediment cores [71], but later [75] refers to 27 sea-ice proxy records. Table 1 lists 24 sediment cores and one ice core, totalling 25 records. So which one is it? Please make this as clear as possible for the frustrated reader.

We have presented 28 sea-ice records from 24 sediment cores (some cores present sea-ice reconstructions through different approaches: both qualitative – *F. curta* group, and quantitative – transfer functions) and an additional record from an ice core. We agree the different numbers given can be confusing. We have provided a sentence to explain this and have made the numbers more consistent throughout the paper.

- How comparable are the quantitative and qualitative reconstructions? With the NaCl in the ice core, would higher values not indicate more open water rather than more sea ice?

The quantitative and qualitative reconstructions in the sediment core records are largely similar in their patterns, as is shown in Figure 2. This is not surprising, as, for the quantitative sea-ice reconstructions, the transfer function output will be strongly (but not only) influenced by the abundance of sea-ice diatoms and so should show strong consistency with the qualitative FCC abundances. The qualitative reconstruction from the EDC ice core is slightly less comparable to the sediment core records, as the $Na_{ss}$ flux record has different sensitivities to sea-ice changes (see review in Thomas et al. (2019)). $Na_{ss}$ flux is also a more integrated signal than marine records as the source area of precipitation/particles reaching EDC, in our case, encompasses the whole Indian Ocean (Delaygue et al. 2000). This is a point that we discuss in lines 195-200.

The sublimation of salty snow from the sea-ice surface is a major source of sea-salt aerosols to Antarctic ice cores, more so than bubble bursting in the open ocean (Yang et al. 2008, Frey et al. 2020). Therefore, increased concentrations of $Na_{ss}$ relate to a greater extent of yearly sea ice.

**Minor comments:**

[lines 26-30]: sea ice is also a crucial habitat for Antarctic organisms, add this information to the paragraph

We have added this information.

[30-35]: model simulations struggle with the internal variability (stochastic nature) of the sea ice system

We have added this information

[40] linked to

We have amended this

[74] data ARE – data = plural

We have amended this

[326] 'heavy sea ice' – please clarify

We have amended this to persistent or long duration to clarify.

Bayer-Giraldi M., Weikusat I., Besir H. & Dieckmann G. 2011. Characterization of an antifreeze protein from the polar diatom Fragilariopsis cylindrus and its relevance in sea ice. *Cryobiology,* **63** (3): 210-219.

Chadwick M., Allen C.S., Sime L.C., Crosta X. & Hillenbrand C.-D. 2022. How does the Southern Ocean palaeoenvironment during Marine Isotope Stage 5e compare to the modern? *Marine Micropaleontology*, **170**: 102066.

Delaygue G., Masson V., Jouzel J., Koster R.D. & Healy R.J. 2000. The origin of Antarctic precipitation: a modelling approach. *Tellus B: Chemical and Physical Meteorology*, **52** (1): 19-36.

Frey M.M., Norris S.J., Brooks I.M., Anderson P.S., Nishimura K., Yang X., Jones A.E., Nerentorp Mastromonaco M.G., Jones D.H. & Wolff E.W. 2020. First direct observation of sea salt aerosol production from blowing snow above sea ice. *Atmospheric Chemistry and Physics*, **20** (4): 2549-2578.

Gersonde R. & Zielinski U. 2000. The reconstruction of late Quaternary Antarctic sea-ice distribution—the use of diatoms as a proxy for sea-ice. *Palaeogeography, Palaeoclimatology, Palaeoecology*, **162**: 263-286.

Janech M.G., Krell A., Mock T., Kang J.-S. & Raymond J.A. 2006. Ice-Binding Proteins from Sea Ice Diatoms (Bacillariophyceae)1. *Journal of Phycology*, **42** (2): 410-416.

Leventer A. 1998. The fate of Antarctic "sea ice diatoms" and their use as paleoenvironmental indicators. In: *Antarctic Sea-ice, Biological Processes, Interactions and Variability*, Lizotte M.P. & Arrigo K.R. Eds., Antarctic Research Series, American Geophysical Union, Washington D.C. **73:** 121-137.

Thomas E.R., Allen C.S., Etourneau J., King A.C.F., Severi M., Winton V.H.L., Mueller J., Crosta X. & Peck V.L. 2019. Antarctic Sea Ice Proxies from Marine and Ice Core Archives Suitable for Reconstructing Sea Ice over the Past 2000 Years. *Geosciences*, **9** (12): 506.

Warnock J.P., Scherer R.P. & Konfirst M.A. 2015. A record of Pleistocene diatom preservation from the Amundsen Sea, West Antarctica with possible implications on silica leakage. *Marine Micropaleontology*, **117**: 40-46.

Xiao W., Esper O. & Gersonde R. 2016. Last Glacial - Holocene climate variability in the Atlantic sector of the Southern Ocean. *Quaternary Science Reviews*, **135**: 115-137.

Yang X., Pyle J.A. & Cox R.A. 2008. Sea salt aerosol production and bromine release: Role of snow on sea ice. *Geophysical Research Letters*, **35** (16).

Zielinski U., Gersonde R., Sieger R. & Futterer D. 1998. Quaternary surface water temperature estimations: Calibration of a diatom transfer function for the Southern Ocean. *Paleoceanography*, **13** (4): 365-383.